# SARS-CoV-2 Alpha-Variant Outbreak Amongst a Partially Vaccinated Long-Term Care Facility Population in The Netherlands—Phylogenetic Analysis and Infection Control Observations

**DOI:** 10.3390/pathogens11101070

**Published:** 2022-09-20

**Authors:** Nathalie Van der Moeren, Veronica A. T. C. Weterings, Suzan D. Pas, Jaco J. Verweij, Wouter van den Bijllaardt, Joyce Geurts, Vivian F. Zwart, Esther B. Lodder, Jan A. J. W. Kluytmans, Jean-Luc Murk, Joep J. J. M. Stohr

**Affiliations:** 1Department of Infection Control, Amphia Hospital, 4818 CK Breda, The Netherlands; 2Microvida, Laboratory of Medical Microbiology and Immunology, Amphia Hospital, 4818 CK Breda, The Netherlands; 3Microvida, Laboratory of Medical Microbiology and Immunology, Elisabeth-TweeSteden Hospital, 5042 AD Tilburg, The Netherlands; 4Thebe, 4813 EC Breda, The Netherlands; 5GGD West-Brabant, 4816 CZ Breda, The Netherlands; 6Julius Center for Health Sciences and Primary Care, University Medical Center Utrecht, Utrecht University, 3584 CG Utrecht, The Netherlands

**Keywords:** SARS-CoV-2, phylogenetic analysis, outbreak

## Abstract

Despite extensive vaccination and booster programs, SARS-CoV-2 outbreaks in long-term care facilities (LTCF) continue to occur. We retrospectively describe a SARS-CoV-2 outbreak amongst a partially vaccinated LTCF population in The Netherlands which occurred in March 2021. The facility comprised three floors functioning as separate wards. Nasopharyngeal swabs for SARS-CoV-2 qRT-PCR were obtained from residents and staff presenting with COVID-19-like symptoms and from all residents and staff during two point prevalence screenings (PPS). Samples meeting technical criteria were included for phylogenetic analysis. Positive SARS-CoV-2 qRT-PCR were obtained from 11 (18%) of 61 residents and 8 (7%) of 110 staff members between March 8 and March 25. Seven (37%) cases and five (63%) vaccinated cases were diagnosed through PPS. Cases were found on all wards. Phylogenetic analysis (*n* = 11) showed a maximum difference of four nucleotides between sequences on the outer branches of the tree, but identified two identical sequences on the root differing maximum two nucleotides from all other sequences, suggesting all did belong to the same cluster. Our results imply that PPS is useful in containing SARS-CoV-2 outbreaks amongst (vaccinated) LTCF populations, as an entire LTCF might behave as a single epidemiological unit and it is preferable to maximize the number of samples included for phylogenetic analysis.

## 1. Introduction

Since the beginning of the epidemic, Coronavrus Disease 2019 (COVID-19) has caused disproportionally high morbidity and mortality amongst residents of long-term care facilities (LTCF). By January 2022, nursing care residents were responsible for 49% of the COVID-19-related deaths in The Netherlands [1]. As a consequence of this augmented risk, LTCF residents have been vaccinated and boostered with priority. Vaccination resulted in a dramatic decrease in severe sickness and death, but outbreaks continue to occur. In August 2022, 440 (18%) Dutch care homes reported Severe acute respiratory syndrome coronavirus 2 (SARS-CoV-2) infections amongst their residents. Basic vaccination coverage in the Netherlands at this moment was 92% amongst individuals 80 years and older; 87% had received a booster vaccine [2].

Elder individuals are known to have impaired immune responses after vaccination as a consequence of an aged immune system [3]. Data on SARS-CoV-2 vaccine effectiveness in this population is scarce, but the available evidence suggests a lower and less durable immune response [4,5,6]. Furthermore, the reduction of hospitalization and case fatality rates in vaccinated LTCF residents was found to be smaller in LTCF outbreaks with the SARS-CoV-2 Delta-variant of concern (VOC) compared to previous variants [4]. Finally, as a consequence of the emergence of new VOC, periods of high levels of SARS-CoV-2 community spread with spillover to LTCF are to be expected. In conclusion, it is clear that vaccination is not the unique golden bullet needed to protect the vulnerable LTCF population from infection with SARS-CoV-2, especially in the light of emerging VOC. Infection prevention and disease control measures will remain an essential part of the protection strategy. Implementation of these measures remains challenging. Even if shortage of skilled staff, scarcity of personal protective equipment (PPE), and lack of access to diagnostics are overcome—protection measures need to be weighed against the quality of life of residents. Existing facilities are often not suited for quarantine and isolation and cognitive disorders amongst residents might interfere with compliance to hygiene measures. Furthermore, in the Netherlands, LTFC are often mixed, facilitating a wide variety of residents from strongly impaired psychogeriatric residents to completely independently living elder couples. The current fragmentation of the organization of medical care for these different types of residents imposes an additional challenge for disease monitoring and control.

In order to provide insights into the course of SARS-CoV-2 outbreaks in LTCF and potential challenges for infection prevention and control, we describe a SARS-CoV-2 Alpha-variant outbreak that occurred in an LTCF in the south of The Netherlands between 8 March and 25 March 2021.

## 2. Materials and Methods

### 2.1. Design

We retrospectively describe an outbreak of SARS-CoV-2 that occurred in a partially vaccinated LTCF population comprising 61 residents with variable care needs and 110 members of staff in the south of The Netherlands between 8 March and 25 March 2021.

### 2.2. Setting

The LTCF was a three-floor building comprising 24 single rooms for psychogeriatric residents (PGR) on the ground floor, 27 apartments for assisted living residents (ALR) on the first floor, and 9 private apartments for independent living residents (ILR) on the top floor. Residents of all floors had access to a communal activity room (CAR) on the ground floor. PGR had private sanitary facilities, but shared a dining and living room per 6 residents. There was an open access connection between two PGR living rooms (Figure 1). Apart from the CAR, ALR and ILR shared no facilities.

PGR required extensive help of staff members for activities of daily living (ADL), ALR received less extensive care from the same group of staff. ILR could benefit from limited help but were generally unaided. PGR received specialized medical care by the facility’s geriatrician; ALR and ILR received care from their own external family practitioner.

In light of the ongoing COVID-19 epidemic in The Netherlands, a number of measures were operative prior to the outbreak: visitors were restricted to a maximum of two per resident per day; visitors were allowed in the residents’ private spaces, but were banned from communal rooms; visitors were registered on arrival and obliged to wear face masks; staff members were obliged to wear surgical face masks and staff with COVID-19-suspect symptoms or high-risk contacts were tested for SARS-CoV-2 by qRT-PCR and furloughed until test results were known; residents witch COVID-19-suspect symptoms or high-risk contacts were tested for SARS-CoV-2 by qRT-PCR and isolated in their room until test results were known; strengthened hygiene practices were in force, including enhanced hand-hygiene measures and frequent disinfection.

At the moment of detection of the first SARS-CoV-2-positive resident on March 8, additional precautions were implemented: all communal activities in the CAR were suspended; PGR residents were restricted to their own living room; all residents who had a positive SARS-CoV-2 real-time reverse transcriptase polymerase chain reaction (qRT-PCR) were isolated in their rooms and nursed with PPE during a 14-day period. Visits were strongly discouraged and restricted to one visitor per client per day and respecting a 1.5 m distance became obligatory.

### 2.3. Participants

Participants were all 61 residents and 110 members of staff living or working at the LTCF on 8 March 2021. Amongst the 61 residents there were 24 PGR, 28 ALR and 9 ILR. Cases were defined as residents or members of staff of whom a positive SARS-CoV-2 qRT-PCR was obtained between 8 March and 15 May 2021. A nasopharyngeal (NP) swab for SARS-CoV-2 qRT-PCR was obtained from all residents and staff presenting with COVID-19-like symptoms such as fever, cough and headache. On 16/17 and 25/26 March, all residents and staff members who had not already been tested positive underwent SARS-CoV-2 qRT-PCR during a facility-wide point prevalence screening (PPS).

The vaccination state of all PGR was obtained from the facility’s geriatrician. Individuals were considered fully vaccinated two weeks after completing a full basic vaccination regimen and partially vaccinated two weeks after receiving the first dose of a multiple dose basic vaccination regimen. All but one PGR were considered fully vaccinated from 12 March on as they had collectively received the second dose of a two-dose regimen two weeks before.

During the outbreak, symptoms and vaccination state of positively tested residents and members of staff were registered in an outbreak log. As ALR, ILR and staff were under medical supervision of their own family practitioner, we did not have access to the vaccination states of the SARS-CoV-2-negative individuals in these groups.

### 2.4. SARS-CoV-2 Real-Time Reverse Transcriptase PCR (qRT-PCR)

All NP swabs obtained by the LTCF were suspended in a 1:1 suspension of lysis buffer and virus transport medium. Nucleic acid extraction and qRT-PCR for SARS-CoV-2 were performed on the Cobas 8800 platform with the CE-IVD labeled Cobas^®^ SARS-CoV-2 PCR assay (Roche, Basel, Switzerland).

Positive SARS-CoV-2 qRT-PCR obtained from four members of staff were obtained by external parties (private laboratories, public health services).

### 2.5. Whole Genome Sequencing (WGS) and Phylogenetic Analysis

Available samples with Ct-values of 30 or less in the E-gene target of the qRT-PCR assay were selected for whole genome sequencing (WGS). Complete SARS-CoV-2 genome sequences were obtained by use of the EasySeq library prep kit (Nimagen, Nijmegen, The Netherlands), sequences were subsequently generated using Miseq next generation sequencer (Illumina, San Diego, CA, USA), fastq files were assembled with CLC Genomics Workbench 20.0.4. Reads were mapped against reference strain NC_045512.2 (Wuhan-Hu-1). Samples with a reference coverage of 70% or more were analyzed using Pangolin (https://cov-lineages.org/, accessed on 9 September 2021) and Nextstrain (https://nextstrain.org/, accessed on 9 September 2021). Samples with a quality of 90% or more were included for alignment and phylogenetic analysis by use of MAFFT and IQ-TREE (https://usegalaxy.org/, accessed on 9 September 2021)) using a maximum likelihood method with bootstrapping (*n* = 1000). Alongside the reference strains, 44 SARS-CoV-2 sequences obtained in the region of the concerned facility during the period of the outbreak were included as background. CLC Genomics Workbench 20.0.4 was used to link the results with the metadata.

### 2.6. Ethics

The PPS was performed to support and guide actual and future infection prevention measures and not for research; as a consequence, no written informed consent was deemed necessary. The LTCF board agreed to the publication.

## 3. Results

A total of 11 residents and 8 staff members tested positive for SARS-CoV-2 between March 8 and 25 March 2021, corresponding to an overall attack rate (AR) of 11%. Amongst the 11 infected residents, there were 5 PGR (AR 21%), 4 ALR (AR 14%) and 2 ILR (AR 22%). One resident died and 1 resident was hospitalized, corresponding to a case fatality rate (CFR) of 9% and hospitalization rate (HR) of 9% (Table 1). In total, 55 residents (90%, positivity ratio 5%) and 105 staff members (95%, positivity ratio 1%) were tested during the first PPS, 52 residents (85%, positivity ratio 4%) and 103 staff members (94%, positivity ratio 1%) were tested during the second PPS. Six (55%) of the 11 qRT-PCR-positive residents were tested because of symptoms, 3 (27%) were detected through PPS on March 16/17 and 2 (18%) through PPS on March 25/26. Amongst the infected staff members, 6 (75%) were tested because of symptoms, 1 member (17%) was detected through the first and one (17%) through the second PPS (Figure 2).

The vaccination state of all but one case was registered. The individual with unknown vaccination state had received one vaccine dose but date and vaccination regimen were unknown. Overall, 8 cases were considered fully vaccinated, 4 were partially vaccinated and 6 were unvaccinated at the time of positive testing. Of the fully vaccinated cases, 5 (63%) were asymptomatic at the moment of diagnosis and detected through PPS compared to 1 (25%) of the partially and 1 (17%) of the unvaccinated individuals (Table 2).

Specimens of all qRT-PCR-positive residents (*n* = 11) and 4 employees (50%) were available at the laboratory. Nine (82%) samples from the residents and 3 (38%) samples from the staff had a Ct-value of 30 or less and were included for whole genome sequencing (WGS). All 12 SARS-CoV-2 genomes detected in the specimens belonged to the B.1.1.7-20I/501Y.v1/UK SARS-CoV-2 lineage. One staff member sample with a reference genome coverage of less than 90% was excluded for subsequent phylogenetic analysis. The viral genomes in the specimen of client 5 and 6 were identical and on the root of the phylogenetic tree. The maximum nucleotide difference between the sequences on the outer branches of the tree was four. However, all included sequences differed a maximum of 2 nucleotides from the 2 sequences on the branch, suggesting all genomes did belong to the same cluster (Figure 3).

## 4. Discussion

We describe a SARS-CoV-2 Alpha-variant cluster amongst 11 residents and 8 employees of a long-term care facility in The Netherlands.

The facility comprised different groups of residents, ranging from completely independent (ILR) to in need of a high level of support (PGR). This is a widely used formula in LTCF in The Netherlands and—as is the case in this facility—the responsibility for medical care of these different groups of residents is often scattered. PGR are often under medical care of a geriatrician attached to the LTCF, but more independent residents tend to have an external family practitioner. Remarkably, SARS-CoV-2 qRT-PCR-positive cases in the herein described outbreak were scattered throughout all floors of the facility and amongst different types of residents early on in the outbreak. Furthermore, phylogenetic analysis showed close associations between residents of different floors. Although the facility is at first sight comprised of distinct floors and groups of residents, it seems to act as one epidemiological unit. It would therefore be logical and recommendable to organize outbreak control and more broadly infection prevention in these facilities in a centralized manner.

Seven (37%) of SARS-CoV-2 qRT-PCR-positive cases were asymptomatic at the moment of diagnosis and detected through point PPS of all residents and employees. As a consequence, 5 residents were isolated and 2 employees were furloughed from work (early), preventing them from further spreading the virus. SARS-CoV-2 is known to be transmitted by pre- and asymptomatic individuals, diagnosis and isolation of these cases are an essential part of outbreak control [7]. Furthermore, SARS-CoV-2 is known to present atypically in the elderly leading to potential delayed recognition and spread [8]. Noteworthy, five (63%) of the fully vaccinated cases were detected through one of these screenings. Facility-wide PPS in context of SARS-CoV-2 outbreaks in LTCF has been shown worthwhile [9]. Our data suggest that these PPS are likely to remain or even become more important for vaccinated LTCF populations. This is likely to be the consequence of the increased proportion of asymptomatic infections amongst SARS-CoV-2-infected vaccinated individuals [10].

Phylogenetic analysis showed a maximum difference of 2 nucleotides between 2 identical variants positioned on the root of the phylogenetic tree and all other sequences. Between variants on the outskirts of the tree however, there was a maximum difference of 4 nucleotides. This number exceeds the 2 SNP per 4 weeks of community mutation rate previously observed by far [11]. A possible explanation would be the analysis of a high number of sequences within a short time frame. However, the majority of genomic epidemiological studies performed in LTCF to date show very little genomic diversity with clusters comprised of identical or near-identical sequences (1 NSP difference) [12]. We hypothesize the epidemic might have started a substantial time before the first case was detected, resulting in a genetic diversity already present at day zero of the outbreak. This is in concordance with what was previously observed in LTCF: by the time two cases are diagnosed, the outbreak is already widespread [12]. Furthermore, the SARS-CoV-2 community mutation rate was determined based on the genetic dynamics of non-variants of concern (VOC) and might be different for the Alpha-variant [11]. The available literature on LTCF is mostly comprised of outbreaks in the first half of 2020 when the non-VOC were circulating in Europe [12,13]. Additionally, based on 2 SNP per 4 weeks of community mutation rate, sequences with a difference of 3 nucleotides or more are commonly considered not to belong to the same cluster. Therefore, if the sequences on the root of the phylogenetic tree would not have been included in the analysis, one could have concluded that not all cases belong to the same cluster, but resulted from multiple introductions. Based on our findings, the inclusion of all available samples when performing phylogenetic analysis in these settings seems recommendable. This is in contrast with a review of 11 genomic epidemiology studies in long-term care facilities (LTCF), where the authors postulate there is no added value of intensive sequencing of all cases and promote the strategic testing of a subset of samples [12].

One of the limitations of the study was the use of the sampling date of a positive test instead of the date of disease onset. The latter would have given a more accurate image of the course of the epidemic, but could not be obtained for a substantial number of individuals. The time between symptom onset and testing, however, was likely to be short as awareness of SARS-CoV-2 was high 12 months into the epidemic in The Netherlands. Furthermore, information on the vaccination state of all ILR and ALR could not be obtained as they were medically managed by their own family practitioner and we had no access to their medical files. As a consequence, no differences in attack rates between vaccinated and unvaccinated individuals could be determined. Furthermore, we had no access to demographic data on residents and employees. The LTCF chose not to share these data because of privacy reasons. Finally, only 3 out of 8 staff member cases could be included for phylogenetic analysis. Strengths of the study are the comprehensiveness and uniformity of testing of residents and employees, and the inclusion of all samples meeting technical quality criteria for WGS and phylogenetic analysis.

In conclusion, our analysis implies that it is advisable to centralize outbreak control in LTCFs, even when the facility is at first sight comprised of distinct floors and groups of residents. In addition, thorough asymptomatic screening of residents and staff members after a SARS-CoV-2 case is detected strongly improves outbreak investigation and is likely to be particularly important amongst vaccinated populations. Having a maximum number of samples included in phylogenetic analysis safeguards against drawing unwarranted conclusions with respect to identifying viral introductions and transmission pathways.

## Figures and Tables

**Figure 1 pathogens-11-01070-f001:**
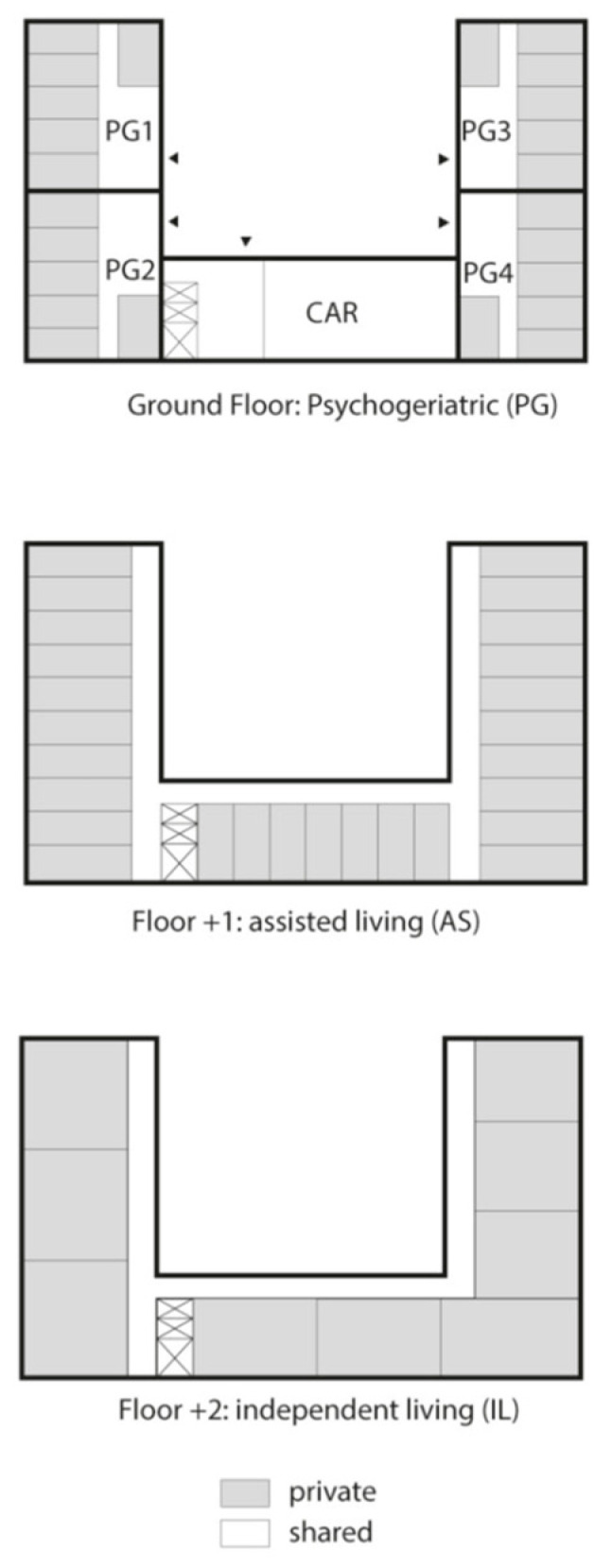
Schematic representation of the long-term care facility floor map with the communal activity room (CAR) situated on the ground floor.

**Figure 2 pathogens-11-01070-f002:**
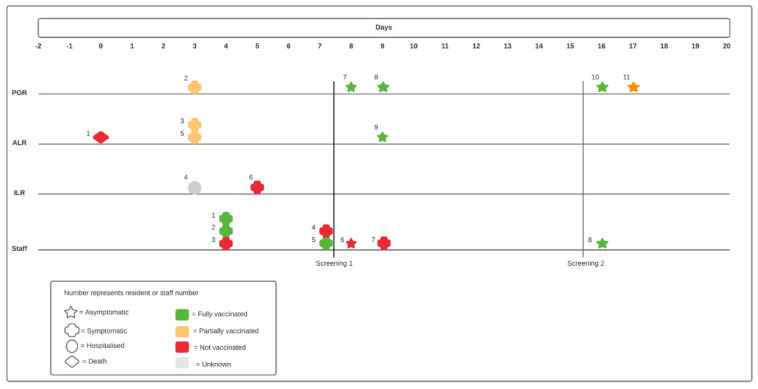
Time line of obtainment of positive SARS-CoV-2 qRT-PCR amongst psychogeriatric (PGR), assisted living (ALR) and independent living residents (ILR), and staff members including clinical image and vaccination state at time of positive qRT-PCR.

**Figure 3 pathogens-11-01070-f003:**
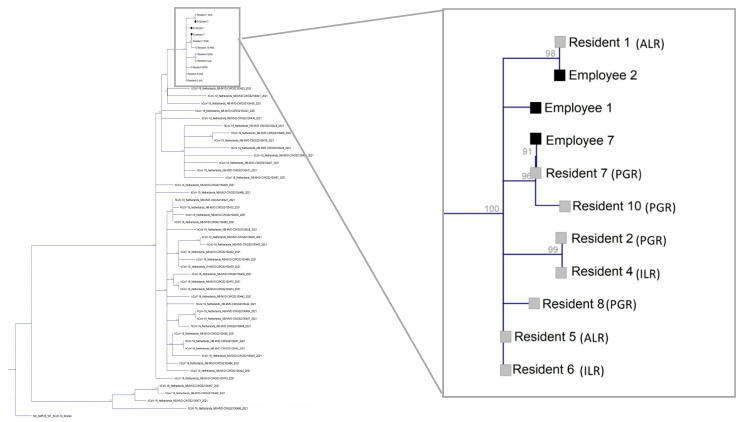
Maximum likelihood tree with bootstrapping (*n* = 1000) of all eligible samples obtained during the outbreak (*n* = 11) and a background of reference strains and 44 SARS-CoV-2 sequences obtained in the region of the concerned facility during the period of the outbreak.

**Table 1 pathogens-11-01070-t001:** Number of cases, case fatality rate, hospitalized cases, symptomatic cases, vaccination status and number of cases detected through point prevalence screening (PPS) stratified by type of resident/staff member.

	Total	Infections *n* (AR%)	Deaths *n* (CFR %)	Hospitalized *n* (HR%)	Symptomatic *n* (%)	Unvaccinated Cases *n* (%)	Partially Vaccinated Cases *n* (%)	Screening *n* (%)
**Total**	171	19 (11%)	1 (1%)	1 (1%)	12 (63%)	6 (32%)	4 (21%)	7 (37%)
**Staff**	110	8 (7%)	0 (0%)	0 (0%)	6 (75%)	4 (50%)	0 (0%)	2 (25%)
**Residents**	**61**	**11 (18%)**	**1 (9%)**	**1 (9%)**	**6 (55%)**	**2 (18%)**	**4 (36%)**	**5 (45%)**
Psychogeriatric (PGR)	24	5 (21%)	0 (0%)	0 (0%)	1 (20%)	0 (0%)	2 (40%)	4 (80%)
Assisted living (ALR)	28	4 (14%)	1 (4%)	0 (0%)	3 (75%)	1 (25%)	2 (50%)	1 (25%)
Independent living (ILR)	9	2 (22%)	0 (0%)	1 (11%)	2 (100%)	1 (50%)	0 (0%)	0 (0%)

CFR: case fatality rate; HR: hospitalization rate; Screening: number of cases detected through point prevalence screening (PPS).

**Table 2 pathogens-11-01070-t002:** Case fatality rate, number of hospitalized cases, asymptomatic cases and number of cases detected through point prevalence screening (PPS) stratified by vaccination status.

	Total	Deaths *n* (CFR %)	Hospitalized *n* (HR%)	Asymptomatic *n* (%)	Screening *n* (%)
**Total number of cases**	19	1 (5%)	1 (5%)	8 (42%)	8 (42%)
**Fully vaccinated**	8	0 (0%)	0 (0%)	5 (63%)	5 (63%)
**Partially vaccinated**	4	0 (0%)	0 (0%)	1 (25%)	1 (25%)
**Unvaccinated**	6	1 (17%)	0 (0%)	1 (17%)	1 (17%)
**Unknown**	1	0 (0%)	1 (100%)	0 (0%)	0 (0%)

CFR: case fatality rate; HR: hospitalization rate; Screening: number of cases detected through point prevalence screening (PPS).

## Data Availability

Not applicable.

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
