# Peer review of "SARS-CoV-2 Alpha-Variant Outbreak Amongst a Partially Vaccinated Long-Term Care Facility Population in The Netherlands—Phylogenetic Analysis and Infection Control Observations"

_pathogens, 2022, doi:10.3390/pathogens11101070_

Round 1
Reviewer 1 Report
Authors did an excellent job working on this article, this article sets an good example for all types of clinical care facilities, With any future outbreaks this kind of preparedness will set high standards. Article mainly focuses on toe point prevalence screenings utility in containing SARS-CoV-2 outbreaks amongst (vaccinated) LTCF populations. I think article can be accepted in its current form after minor corrections.
Minor Comments:
1. Typographical errors
2. Resolution of figures can be improved for better understanding.
Author Response
Dear reviewer,
Thank you for your careful review and constructive suggestions.
Please find our answers to your remarks below in italics.
Yours sincerely,
Nathalie Van der Moeren
In name of the co-authors
- Typographical errors
Thank you for this remark, we corrected a number of typographical errors. Amongst others we uniformized the text in US English.
- Resolution of figures can be improved for better understanding.
Thank you for this important remark, the figures were uploaded in pdf alongside with the revised manuscript in order to be inserted in the manuscript in the highest possible resolution.

Reviewer 2 Report
Overall this is a comprehensive manuscript, with clear messages that are however not as clearly addressed as they could be.
Also at this stage, within continuous virus evolution and emergence of new VOCs with potential risks of immune escape the subject remains relevant.
Although overall the manuscript is of interest for publication, there are a number of recommendations for improvement, which are listed below.

Author Response
Dear reviewer,
Thank you for your careful review and constructive suggestions. We feel the manuscript is substantially improved after making the suggested edits.
Please find our answers to your remarks below in italics.
Yours sincerely,
Nathalie Van der Moeren
In name of the co-authors
General comment
- Although the manuscript is generally well written, the use of broken/ in between sentences ( -text -) should be limited, as it makes complicates fluent reading.
Thank you for this remark, adjustments were made and broken sentences were removed from the manuscript. (Line 49 and 108)
- The most important limitation in the study is the absence of a clear overview of the study population, including demographics and vaccination coverage in the different populations and staff within the facility. To better appreciate the data and conclusions it is important that such data are presented. This also reflects the presentation of the results .
Thank you for this remark, unfortunately we didn’t have access to demographic data of residents and employees. The LTCF management was not able to/ willing to share these data because of privacy reasons (Dutch privacy law).
As vaccination status of non-PGR residents and employees was not known to the LTCF management (for privacy reasons) this data could not be included.
We elaborated on this issue in the limitations section of the reviewed conclusion. (Line 267 -273)
Introduction
- The context provided is very much oriented on the EU situation, whereas the study concerns a Dutch LTCF. A description of the epidemiological context of this particular setting is missed and should be given more attention. The reference to the ECDC figures, is too generic and does not provide a sufficient context for this study.
Thank you for this valuable remark, we focussed the context on the Netherlands, ECDC figures were replaced by figures of the Dutch RIVM. (Line 41-44) - The text … as a consequence of incomplete general vaccination uptake… (line 51) should be clarified in relation to the above.
We decided to delete this part of the sentence as no figures on the vaccination uptake amongst Dutch healthcare workers are available. (Line 55)
- VOC should once again be explained when first used.
An explanation of the abbreviation VOC was added to the manuscript. (Line 54)
Materials and Methods
- Line 122 mentions that no information was available on vaccination state of SARS-CoV-2 negative ALR, ILR and staff members. This should be explained, both in relation to the sentence in line 51, which suggest a better insight and the context of the results presented.
Thank you for this remark, The explanation ‘As ALR, ILR and staff were under medical supervision of their own family practitioner, we did not have access to the vaccination states of SARS-CoV-2 negative individuals in these groups.’ Was added to the manuscript. (Line 130-133)
Results
The results description requires improvement.
- The subjects tested through PPS comprise 56 residents and 104 staff If so this should be mentioned, and screening data should be calculated against that figure.
These figures divided in first and second PPS (including positivity rates) were added to the manuscript. (Line 164-167)
- It recommended to include a table with a more detailed demographic description, including the vaccination status.
Thank you for this remark, unfortunately we didn’t have access to demographic data of residents and employees. The LTCF management was not able to/ willing to share these data because of privacy reasons (Dutch privacy law). The lack of these data was addressed in het limitations section. (Line 267 -273)
- Line 157- mentions that one staff member tested positive twice and is counted twice. Given the short interval between the 2 screening dates it questionable whether this should be counted as 2 events.
Thank you for this important remark, the formulation of this sentence was misguiding. There was one staff member detected through the first and one through the second PPS. No one was tested twice. The sentence was adapted in the manuscript. (Line 170-171)
- Tables -2. Denominators are different for the different tables, resulting in percentages that may not be comparable, e.g., CFR of total cases is apparently based upon al subjects (n=171), but in the unvaccinated in the known unvaccinated population (n=6). This is not correct, as the vaccination status of the non-positives is not provided. The tables should be adapted accordingly, or rates should be deleted.
We are very thankful for this correction, we corrected the table so al denominators are the total number of cases in the specific category to correctly depict case fatality and hospitalisation rates in these different categories. (Table 2)
- Although the authors mention that facility wide PPS and infection prevention is important, the manuscript does not describe the specific measures within the timeline and how they lead to the ending the outbreak.
Thank you for this valuable remark, we added the specific measures taken in case of a positive qRT-PCR to the Methods section. (Line 108-109) Furthermore, we elaborated on the subject in the discussion. (Line 226-228)
Discussion
- The discussion is clearly written, although rather extensive on the sequence discussion at the expense of the discussion on the other recommendations mentioned in the abstract, namely that PPS is useful in containing SARS-CoV-2 outbreaks amongst (vaccinated) LTCF populations, and entire LTCF might behave as a single epidemiological unit.
Thank you for this remark, we elaborated on the missing demographic and vaccination data of PCR negative residents and staff. (Line 269-273) Furthermore, we added the potential atypical presentation of SARS-CoV-2 amongst the elderly to the discussion on relevance of PPS in these populations. (Line 226-231)
- It furthermore recommended to give consideration to several of the points raised above, including the description of the study population, the absence of any information on the test negatives (unless provided in the response), a discussion on the role and effects of the infection prevention measures.
Thank you for these important remarks, the first items were added to the limitation section. The infection prevention measures were elaborated on in line 226-228.
- Line 245: One of the limitations of the study was the use of the sampling date of a positive test instead of the date of disease onset. Should be further explained. It suggests no adequate insight into the clinical presentation of the patients, which should not be the case. What was the daily practice, and recording for clinical follow up of the residents?
Thank you for this remark. Unfortunately the exact moment of disease onset was not registered for all individuals. It can be challenging to determine when a cognitively impaired elderly developed a sour throat or a limited general malaise, on the moment there were clear symptoms or cognitively adequate residents had complaints a sample for PCR was immediately obtained as awareness so far in the epidemic was high as stated in the text. We do realize however this is a limitation.
- It is recommended to critically review the discussion following changes made to the manuscript.
We believe we did so based on the reviewers remarks and believe it strengthens our manuscript, for which we are grateful.
